# A 69 kbp Deletion at the Berry Color Locus Is Responsible for Berry Color Recovery in *Vitis vinifera* L. Cultivar ‘Riesling Rot’

**DOI:** 10.3390/ijms23073708

**Published:** 2022-03-28

**Authors:** Franco Röckel, Carina Moock, Florian Schwander, Erika Maul, Reinhard Töpfer, Ludger Hausmann

**Affiliations:** Julius Kühn Institute (JKI), Institute for Grapevine Breeding Geilweilerhof, 76833 Siebeldingen, Germany; carina.moock@julius-kuehn.de (C.M.); florian.schwander@julius-kuehn.de (F.S.); erika.maul@julius-kuehn.de (E.M.); reinhard.toepfer@julius-kuehn.de (R.T.)

**Keywords:** anthocyanin, BAC, grapevine, homologous recombination, MYB transcription factor, pedigree analysis, qRT-PCR

## Abstract

‘Riesling Weiss’ is a white grapevine variety famous worldwide for fruity wines with higher acidity. Hardly known is ‘Riesling Rot’, a red-berried variant of ‘Riesling Weiss’ that disappeared from commercial cultivation but has increased in awareness in the last decades. The question arises of which variant, white or red, is the original and, consequently, which cultivar is the true ancestor. Sequencing the berry color locus of ‘Riesling Rot’ revealed a new *VvmybA* gene variant in one of the two haplophases called *VvmybA3/1RR*. The allele displays homologous recombination of *VvmybA3* and *VvmybA1* with a deletion of about 69 kbp between both genes that restores *VvmybA1* transcripts. Furthermore, analysis of ‘Riesling Weiss’, ‘Riesling Rot’, and the ancestor ‘Heunisch Weiss’ along chromosome 2 using SSR (simple sequence repeat) markers elucidated that the haplophase of ‘Riesling Weiss’ was inherited from the white-berried parent variety ‘Heunisch Weiss’. Since no color mutants of ‘Heunisch Weiss’ are described that could have served as allele donors, we concluded that, in contrast to the public opinion, ‘Riesling Rot’ resulted from a mutational event in ‘Riesling Weiss’ and not vice versa.

## 1. Introduction

Plants, in general, produce more than 200,000 different metabolites, many of which are pigmented [1]. One of the main functions of plant pigments, which play a central role in plant evolution, is the coloring of fruits and flowers for interaction with pollinators and seed dispersers [2]. Aside from chlorophyll, there are three major groups of plant pigments: carotenoids, betalains, and anthocyanins, which can range in color from yellow/orange to blue/purple. The anthocyanins (from the Greek “anthos” = flower and “kyanos” = blue) occur ubiquitously in nature [3] and belong to a very large group of secondary plant metabolites, the flavonoids, of which more than 4000 known compounds have been isolated [4]. Anthocyanins are soluble in both alcohol and water and have a strongly pH-dependent color spectrum. The flavonoid biosynthetic pathway in grapevine splits into three branches leading to the formation of three groups, the flavonols, the proanthocyanidins, and the anthocyanins [5]. The starting products of all flavonoids and consequently of the anthocyanins is malonyl-CoA (malonyl-coenzyme A) and the activated cinnamic acid derivative p-coumaroyl-CoA (p-coumaroyl-coenzyme A) [6]. In the last step of the pathway, after glycosylation by UFGT (UDP-glucose:flavonoid-3-O-glucosyltransferase), the intensely colored anthocyanins (all precursors are colorless) are formed and possibly further methylated. Theoretically, around 100 possible anthocyanins could be found in grapevines. Within the genus *Vitis*, pelargonidin derivatives could only be detected from descendants of *Vitis amurensis* or *Vitis labrusca*. The anthocyanins of our cultivated grapevine varieties are made up of the five anthocyanidins cyanidin, peonidin, delphinidin, petunidin, and malvidin [7,8]. One of the most important external factors for the formation and quantity of anthocyanins in grapevine is light, but too high temperatures and too much light can have a negative impact on anthocyanin production [9].

The ancestors of today’s cultivated grapevine cultivars, as well as almost all wild *Vitis* species, have blue/black berries at maturity ([10]; https://www.vivc.de/ accessed on: 28 February 2022). The origin of white varieties is believed to be in the Middle East, as analysis of the grave goods of the Egyptian pharaoh Tutankhamun (1332–1322 BC) showed that the Egyptians were already making white wine more than 3000 years ago [11]. White varieties do not produce any anthocyanins in berry skins during ripening because, unlike colored grape varieties, they cannot express *VvUFGT* [12]. A single locus on chromosome 2 (at approximately 14.2 Mb relative to the PN40024 12x reference genome [13]), which is called the grapevine berry color locus (BCL), contains multiple *VvmybA* genes of the R2R3 type and controls *VvUFGT* expression in colored berries during ripening [14]. *VvmybA1* and *VvmybA2* are dominant genes, functional in red cultivars, and differ only by a C-terminal domain duplicated in *VvmybA2*. In addition, this locus contains another *VvmybA* gene (*VvmybA3*) with a complete sequence, which, however, is not functional in colored grape varieties due to a premature stop codon in the open reading frame (ORF). The two originally functional genes *VvmbA1* and *VvmybA2* are mutated in white varieties. In the case of *VvmybA1*, expression is prevented by the retrotransposon *Gret1*, which is inserted into the promoter region. According to estimates by Mitani et al. [15] the insertion of *Gret1* could have happened around 110,000 to 290,000 years ago. In contrast, two amino acid sequence altering mutations within the ORF of *VvmybA2* led to a structural change of the protein and consequently to non-functionality [14]. The first mutation is located in the DNA-binding N-terminal region of the R2R3 domain, whereas the second mutation in the C-terminal region (2 bp deletion) leads to a stop codon and thus to a shortened protein sequence. In combination, both mutated *VvmybA* variants led to the formation of the white allele, which is homozygous in almost all white-berried varieties [16]. Because most berry color mutations can easily be detected in the vineyard, a large number of cultivar clones with different colors have been selected since the rise of viticulture. Most gain of function mutations led to a color reversion from white to red, however, this type of mutation never resulted in a blue/black berry color. A possible reason is that the color recovery mutations from white to red due to recombination or rearrangement events within the *VvmybA* genes of the BCL regularly resulted in only one functional gene. This means a second functional *VvmybA* gene, which is present in the traditional red wine varieties with black berry color, is absent in the color mutants [17].

The first mention of the white grapevine variety ‘Riesling Weiss’ dates back to a winery invoice from the Counts of Katzenelnbogen in Rüsselsheim, Germany, in the year 1435 [18]. As a direct descendant of ‘Heunisch Weiss’ and probably a seedling of ‘Savagnin Blanc’ and *Vitis sylvestris* [19], it is generally assumed that the cradle of ‘Riesling’ lies in the Rhine Valley between Karlsruhe and Worms in Germany [20]. With more than 20,000 hectares of area under cultivation (https://www.vivc.de/ accessed on: 28 February 2022), ‘Riesling Weiss’ is the most planted grapevine variety in Germany and is furthermore world-famous for fresh and fruity wines high in acidity. On the other hand, nothing is known about the origin of the red-berried variant ‘Riesling Rot’, but it can be assumed that it was already planted in mixed plots of the late Middle Ages. Presumably, it was a rare cultivar variant before the phylloxera crisis, so after the switch to varietal vineyards, ‘Riesling Rot’ disappeared completely from cultivation for unknown reasons. Similar to many other varieties without a practically relevant area under cultivation, ‘Riesling Rot’ survived in German grapevine repositories where clone selection began in Geisenheim, Germany in 1991. Since 2002, increased cultivation started and Germany-wide classification has meanwhile been achieved. Ampelographically, ‘Riesling Rot’ can only be distinguished from ‘Riesling Weiss’ by the red berry color (Figure 1). In contrast to most color mutants of other grape varieties, however, the color is not completely stable, so occasionally white grapes can be found on a red vine (Figure 1C). Since this phenomenon could be observed multiple times by winegrowers, but never the mutation from white to red, it was suggested that ‘Riesling Rot’ represents the original form and ‘Riesling Weiss’ may have arisen from it. ‘Riesling Rot’ was titled as the true, long-forgotten ancestor of the famous ‘Riesling Weiss’, raising public interest and supporting commercialization tremendously. However, the hypothesis could never be proven, and based on the scientific knowledge about color mutants of grapevine, the assumption seems implausible.

The main objective of this study was to identify the specific mutation leading to berry color recovery in ‘Riesling Rot’ and to draw conclusions to the pedigree of ‘Riesling Weiss’. Due to a large deletion between *VvmybA1* and *VvmybA3*, we sequenced the complete berry color locus of the mutated haplophase of ‘Riesling Rot’ via a bacterial artificial chromosome (BAC) library. Furthermore, we created and genotyped berry color locus-specific homozygotes via self-crossing of ‘Riesling’ accessions and determined the inheritance of the mutation-carrying haplophase of ‘Riesling Rot’. Based on these results, we propose a pedigree tree of ‘Riesling Weiss’ and ‘Riesling Rot’ incorporating the inheritance of the haplophase linked to the specific mutation.

## 2. Results

### 2.1. Creation of ‘Riesling’ Selfing Lines Homozygous at the Berry Color Locus

In previous studies on color mutants of white grape varieties, *VvmybA*-related mutations leading to berry color recovery during ripening were described [21,22,23]. However, many color clones were either not examined or the specific mutation could not be identified. A possible reason is a heterozygosity at the berry color locus, with the non-mutated white haplophase interfering, especially in PCR analyses. To allow a clear haplophase-specific molecular analysis, ‘Riesling Weiss’ (RW; homozygous for the white allele), the color variant ‘Riesling Rot’ (RR; heterozygous), and a revertant of ‘Riesling Rot’, where the red berry color mutated back to white (BRR; again homozygous for the white allele), were self-crossed by means of open pollination followed by a selection of seedlings based on their haplotype constitution at the BCL (Appendix A). In a first step, about 90 seedlings per self-crossed accession were genotyped with nine genome-wide SSR markers ([24], https://www.vivc.de/ accessed on: 28 February 2022) to confirm true self-pollination (Appendix A). A seedling was considered as a true self-cross if only parental alleles could be observed for the nine genome-wide SSR markers. In a second step, six additional SSR markers on each side of the BCL (VMC6B11, VMC5G7, GF02-55, GF02-50, VMC8C2, and VMC7G3) were applied to differentiate haplophases A (later identified as the haplophase without mutation in ‘Riesling Rot’) and B (later identified as the haplophase with a mutation in ‘Riesling Rot’). Six RWs (self-crossed seedlings of ‘Riesling Weiss’), twelve RRs (self-crossed seedlings of ‘Riesling Rot’), and eight BRRs (self-crossed seedlings of the RR revertant to white) seedlings were selected based on different BCL-specific SSR marker profiles that revealed homozygosity (A/A or B/B) or heterozygosity (A/B) along chromosome 2. Since there is a strong inbreeding depression in grapevines [25], special attention was paid to healthy and vigorous growth during seedling selection.

### 2.2. The Unique Mutation of ‘Riesling Rot’

In the beginning, ‘Riesling Rot’ was examined for mutations in the *VvmybA* genes already known from the literature [17,26], but no mutation could be identified. However, in the course of the analysis of the homozygous self-cross line RRs with the SSR markers located in the BCL, a genomic region could be identified as missing in seedlings RRs9-12 homozygous for haplophase B, but not in seedlings RRs1-8 and seedlings RWs3-6 of ‘Riesling Weiss’ with the corresponding haplophase (Appendix A). All four markers (GF02-68, GF02-69, GF02-70 and GF02-72) located between *VvmybA1* and *VvmybA3* did not amplify. Since ‘Riesling Rot’ and seedlings RRs5-8 are heterozygous, the loss of the respective region could not be detected due to the presence of the non-mutated haplophase A. In contrast, the missing genomic region could be detected in the seedlings BRRs5-8 of the ‘Riesling Rot’ revertant mutant homozygous for haplophase B. It can therefore be assumed that the original haplotype constitution at the BCL was restored by a reversion mutation from ‘Riesling Rot’ to white.

In order to display the complete mutated allelic haplophase B at the sequence level, a BAC library for ‘Riesling Rot’ was created, and BAC clones with the corresponding haplophases A (BAC07A05; not mutated) and B (BAC05I18; mutated) were sequenced with Illumina MiSeq (2 × 250 bp paired-end reads, approximately 100× coverage). A schematic overview of both sequenced BACs with annotated *VvmybA* genes (Appendix A) is given in Appendix A (complete sequences deposited at GenBank database under OM885364 and OM885363). The sequencing of the “white” BAC07A05 showed no significant differences compared to the reference sequence PN40024 (12×). Two minor differences in the lengths of repetitive elements that could have been caused by processing the Illumina raw data were not considered further. In contrast, the sequencing of haplophase B (BAC05I18) from ‘Riesling Rot’ revealed a new, recombinant *mybA* gene variant which was named *VvmybA3/1RR* (size: 2097 bp) based on its structure (Figure 2A). Due to the complete recombination of *VvmybA1* and *VvmybA3* with an overlap of 109 bp that cannot be clearly assigned, the allelic variant differs from the original *VvmybA1* ORF only by a single point mutation and the subsequent predicted protein sequence by one amino acid (I12T) (Appendix A). However, the promoter region originates from *VvmybA3,* and the complete region between *VvmybA1* and *VvmybA3,* including *Gret1* (~69 kbp), was deleted on haplophase B of ‘Riesling Rot’. To validate the results from the BAC clone sequencing for all ‘Riesling’ accessions and self-cross lines, a PCR assay was applied to confirm the presence or absence of the *VvmybA3/1RR* allele as well as *VvmybA1* and *VvmybA3* (Figure 2B; Appendix A). Furthermore, the *VvmybA3/1RR* allele could additionally be verified for all 16 independent ‘Riesling Rot’ accessions from German repositories and maintained in vineyards of winegrowers (Appendix A).

### 2.3. The Anthocyanin Concentration and Expression of the VvmybA3/1RR Allele Is Related to the Haplotype Constitution

The visual inspection of the greenhouse grown self-cross line of ‘Riesling Rot’ exhibited enhanced dark red/blue berry coloration of the seedlings RRs10 and RRs11 homozygous for the mutated allele in comparison to the red-berried heterozygous seedlings RRs5 and RRs6 (Figure 3A). Subsequent HPLC anthocyanin analysis of berry skin samples confirmed the first impression. The average berry skin anthocyanin content was increased in RRs10 and RRs11, and RRs11 even reached the level of the red wine reference cultivar ‘Pinot Noir’ with around 5 mg/g total anthocyanins in the skin dry weight (Figure 3B). RRs10 noticeably exceeded with 2.1 mg/g total anthocyanins RRs5 and RRs6, but did not reach RRs11 or ‘Pinot Noir’. RRs5 and RRs6 showed the level of ‘Riesling Rot’ with around 1 mg/g total anthocyanins. Additionally, the ratio of the two main anthocyanin groups was significantly shifted from 3′-hydroxylated anthocyanins towards 3′,5′-hydroxylated anthocyanins from approximately 90:10 in the heterozygous seedlings RRs5 and RRs6 to around 50:50 in the homozygous seedlings RRs10 and RRs11 (Appendix A). 

The same ‘Riesling’ samples were used to determine and confirm the relative expression of *VvmybA1*/*VvmybA3/1RR* and *VvmybA3* using qRT-PCR. Since the wild type *VvmybA1* ORF is, except for one SNP at the N-terminus (Appendix A), identical to the *VvmybA3/1RR* variant, the primer combination used for expression analysis cannot distinguish between both genes in practical analysis. Therefore, ‘Riesling Weiss’ was additionally analyzed as a negative control to confirm the missing *VvmybA1* expression, although already Kobayashi et al. [27] demonstrated that *VvmybA1* is not expressed in white cultivars. In all analyzed samples, expression of *VvmybA3/1RR* and *VvmybA3* reflected the differences in haplotype constitution at the BCL. The average expression levels of *VvmybA3/1RR* were 80-100-fold higher in the homozygous seedlings RRs10 and RRs11 with two gene copies compared to the heterozygous seedlings RRs5 and RRs6 with one gene copy (Figure 3C). Furthermore, expression of *VvmybA3* could only be detected in RRs5 and RRs6 and not in RRs10 and RRs11, and the expression in ‘Riesling Weiss’ was increased two to three-fold compared to ‘Riesling Rot’ that is in relation to the differences in gene copy number due to the mutated ‘Riesling Rot’ allele (Figure 3D).

### 2.4. ‘Riesling Rot’ Is a Mutant of ‘Riesling Weiss’ and Not Vice Versa

A key question with increased public interest regarding ‘Riesling Weiss’ and ‘Riesling Rot’ is the direction of the mutation from white to red or from red to white. Whereas for recently bred cultivars like ‘Kerner’ and ‘Mueller Thurgau Weiss’, color recovery mutations could be observed and documented (https://www.vivc.de/ accessed on: 28 February 2022); the first occurrence of ‘Riesling Rot’ is not well documented and assumed to be centuries old. Furthermore, due to the frequently occurring back mutations from red to white and the missing observation of a mutation from white to red, the direction of inheritance was questionable and winegrowers speculated that ‘Riesling Rot’ may be the progenitor. For this purpose, the SSRs, which were genotyped for the screening regarding homozygosity at the BCL, were also analyzed in the known white-berried ‘Riesling’ parent ‘Heunisch Weiss’. Based on the BCL-specific genetic profile of ‘Heunisch Weiss’, the inherited ‘Riesling Rot’ haplophases could be assigned to the respective parents (Appendix A). It became evident that the mutation-bearing haplophase (possesses the *VvmybA3/1RR* allele) originates from the haplophase originally passed on by ‘Heunisch Weiss’. Consequently, the white variant ‘Riesling Weiss’ must have been first and mutated to red since no color mutant is described for the parent variety ‘Heunisch Weiss’ that could have passed the mutation. A complete pedigree of ‘Riesling’ based on the results of the haplophase assignment is shown in Figure 4. 

## 3. Discussion

### 3.1. Putative Molecular Cause of the VvmybA3/1RR Allele in ‘Riesling Rot’

The plethora of naturally occurring color mutants helped researchers to elucidate the anthocyanin biosynthetic pathway via analysis of gain-of-function and loss-of-function mutations. Initial studies of the anthocyanin biosynthetic pathway in grapevines showed that the difference between white and red varieties is probably due to the loss of *VvUFGT* expression [12]. Another study by Kobayashi et al. [28], which included a detailed sequence analysis of *VvUFGT* in different cultivars, resulted in no significant differences in white and colored cultivars and it was suggested that mutated MYB transcription factors were probably the molecular cause for the differences in grape skin coloration. The same working group was then able to show that the insertion of the retrotransposon *Gret1* into the promoter region of *VvmybA1* is associated with the loss of color formation in white grape varieties by preventing the *VvmybA1* expression [27]. In the same work, an allele called *MybA1b* could be described for ‘Ruby Okuyama’ (color mutant of ‘Italia blanc’) and ‘Flame Muscat’ (color mutant of ‘Muscat of Alexandria’). The sequence of *Gret1* in front of *VvmybA1* was deleted and only a single LTR surrounded by a 5 bp duplicated target site remained, leading to color recovery in the respective cultivars. Originally, the ABRE-like sequences (ABA-Responsive Element) probably required for the transcription of *VvmybA1* were separated by *Gret1*, but due to the loss (excision of *Gret1*), they are again in close proximity and putatively enable transcription [21]. A model that can explain the occurrence of individual LTRs in the genome was described by Puchta [29]. Thus, due to further active transposons [30] or erroneous integration attempts of other retrotransposons [31], a double-strand break (DSB) could have occurred and was then repaired by homologous recombination. In general, this mutation (*MybA1b* allele) seems to be the most common, as it has already been described for many grapevine color mutants [22,23,32]. However, since previous studies have mainly focused on the analysis of *VvmybA1* and *Gret1* [17,22,23], it can be assumed that some mutations have certainly remained undetected.

Several mechanisms of homologous recombination have been described for plant somatic tissue, two of which are considered particularly important. In addition to the synthesis-dependent strand annealing mechanism (SDSA), which is one of the conservative mechanisms (no sequence information is lost), the non-conservative (sequence information is lost) single-strand annealing mechanism (SSA) plays an important role [33,34]. In the first step after a DSB, the 5′ ends are resected leading to two free 3′ overhangs. If homologous sequences are now available within both free 3′ ends, hybridization can occur. Gaps in the DNA backbone are closed and the overhanging 3′ ends are removed. The complete sequences between the homologous areas as well as one of the two sequence repetitions are excised. SSA plays a superior role in genomic regions with sequence repeats that are in close proximity [35], but it has also been shown that deletion by SSA can take place despite larger distances between the repeats [36]. An example of this is putatively the already described *MybA1b* allele and the *VvmybA3/1RR* allele of ‘Riesling Rot’ described in this study since the SSR marker analysis and the BAC clone sequencing of the mutated haplophase revealed the loss of the complete genomic region between *VvmybA1* and *VvmybA3*. A DSB probably occurred in the area in between, possibly also in *Gret1*, which in this case led to homologous recombination of *VvmybA1* and *VvmybA3* through SSA and resulted in the loss of approximately 69 kbp of sequence information. Furthermore, since the mutation could be detected in 16 further independent ‘Riesling Rot’ accessions across Germany, a single mutational event must be assumed. 

### 3.2. Back Mutations to White and Ancestry of ‘Riesling Rot’

In contrast to most other color mutants, a back mutation of ‘Riesling Rot’ to white berry color (no detectable anthocyanins in the berry skins during ripening) occurs rather frequently. A mutated shoot of ‘Riesling Rot’ was cut in winter after seasonal lignification and propagated, leading to a ‘Riesling’ vine with stable white berries. A self-cross line selected for homozygosity of the berry color locus haplophases was created and it could be proven by SSR marker analysis that the region missing between *VvmybA1* and *VvmybA3* in haplophase B of ‘Riesling Rot’, seems to be completely present again in the revertant (Appendix A). It has to be assumed that this genomic region is completely derived from the homologous chromosome (haplophase A; not mutated) of ‘Riesling Rot’. It can furthermore be suggested that the SDSA mechanism and the double-strand break repair model (DSBR) are putative causes [37,38,39]. The phenomenon of interchromosomal homologous recombination has already been described, for example, for *Nicotinia tabacum* [40], *Zea mays* [41], or *Arabidopsis thaliana* [42] and plays a crucial role in DNA repair in somatic cells. Furthermore, since usually only a few terminal bunches or only certain parts of a bunch are mutated in ‘Riesling Rot’, it can be supposed that the back mutations arise during apical meristem bud development [43]. Comparable back mutations from red to white are also known from the two-color mutants of the cultivar ‘Italia blanc’ (‘Ruby Okuyama’ and ‘Benitaka’) which have two different color recovery mutations [44]. Therefore, the occurrence of back mutations seems variety-specific rather than mutation-specific and Collet [44] already assumed that the retrotransposon *Gret1* upstream of *VvmybA1* may be related to the back mutation, as retrotransposons, in general, represent hotspots for recombination [45,46]. In contrast, *Gret1* is no longer present in the *VvmybA3/1RR* allele of ‘Riesling Rot’ (Figure 2A), but it can be suggested that due to the loss of approximately 69 kbp of sequence information and the associated restructuring of the locus, changes in DNA topology or epigenetic modifications could play a crucial role. Since the phenomenon of back mutation to white could be observed multiple times by winegrowers, but not the initial mutation from white to red, the assumption that ‘Riesling Rot’ represents the original clone and ‘Riesling Weiss’ is the descendant was distributed in German-speaking countries. In addition, because ‘Riesling Rot’ was not present in commercial viticulture until the last decades, the so-called rediscovery of the real ‘Riesling Weiss’ ancestor led to a marketing boost with the increased public interest. However, based on the homozygous selfing lines of ‘Riesling Rot’ determined by the SSR marker profile along chromosome 2, the detected mutation leads to berry color formation that could be assigned to the haplophase inherited by the known white-colored parent ‘Heunisch Weiss’ (Figure 4; Appendix A). No true color mutant has been described in the literature for ‘Heunisch Weiss’ that could have served as progenitor and subsequent donor of the mutation. Therefore, it must be concluded that the haplophase was passed on from ‘Heunisch Weiss’ to ‘Riesling Weiss’ and the mutation to red took place in ‘Riesling Weiss’. If, for example, in ‘Riesling Rot’ the mutated haplophase would have originated from the second unknown parent, it could not be excluded that the mutation would have been inherited and consequently ‘Riesling Rot’ would have been first. Further proof would be to demonstrate that the resulting revertants of ‘Riesling Rot’ are genetically different from the original ‘Riesling Weiss’. Since the entire genomic region between *VvmybA1* and *VvmybA3* is missing on the mutated haplophase in ‘Riesling Rot’, but is then restored via the homologous chromosome in the back mutants, the sequence could putatively possess differences in comparison to the original haplophase from ‘Riesling Weiss’. By sequencing the corresponding region and identifying mutations in ‘Riesling Weiss’, clear evidence could be provided that the resulting back mutants differ genetically from the original variety. However, no haplophase-specific mutation could be found in the region between *VvmybA1* and *VvmybA3* in ‘Riesling Weiss’ that could have served as starting point for further analysis (data not shown; sequence information for both ‘Riesling Weiss’ haplophases kindly provided by Camille Rustenholz, University of Strasbourg, Strasbourg, France). 

## 4. Materials and Methods

### 4.1. Plant Material and Sampling

A complete plant list of materials with specific utilization is shown in Appendix A. ‘Riesling’ accessions and reference cultivars are planted as living specimens in the grapevine genbank of the Julius Kühn Institute (JKI)—Institute for Grapevine Breeding Geilweilerhof, Germany. Self-cross lines were created in 2013 from ‘Riesling Weiss’ (*V*IVC variety no. 10077) and ‘Riesling Rot’ (*V*IVC variety no. 10076) and 2014 (back mutation of ‘Riesling Rot’ to white) by open pollination. Grape bunches were harvested after seed maturation in September the seeds were stored and seeded in small pots in March of the following year, and, after selection of the suited genotypes, planted in soil in the greenhouse for further use.

Berry samples of the varieties were taken from three independent field-grown vines in 2015 and of seedlings from one greenhouse-grown vine in 2020 (first flowering) at maturity (at least 16° Brix). Ten ripe berries of three sun-exposed bunches were collected. Berry skins were separated, weighed, freeze-dried, and stored in the dark until further use. The samples for transcript analysis were immediately frozen in liquid nitrogen and stored at −80 °C. Anthocyanin analysis and quantitative real-time PCR (qRT-PCR) transcript analysis for *VvmybA* genes (primers listed in Appendix A) were performed as described in [47]. The Student’s t-test and Tukey’s HSD test of the R-software environment [48] were used to determine statistically significant differences of anthocyanin and gene expression data. 

### 4.2. DNA Extraction and SSR Marker Analysis

DNA was extracted from young leaves using the Plant DNA Mini Kit (Peqlab, Erlangen, Germany) according to the supplier’s instructions. The plant material was genotyped with nine genome-wide SSR markers to confirm trueness-to-type and successful selfing ([24], https://www.vivc.de/ accessed on: 28 February 2022). Six additional, for ‘Riesling’ polymorphic, SSR markers on chromosome 2 (VMC6B11, VMC5G7, GF02-55, GF02-50, VMC8C2, and VMC7G3) were used to determine the haplophase type and their inheritance in the pedigree of ‘Riesling’. Marker genotyping was conducted as described in [49]. A complete list of the BCL-specific SSR markers used in this study is shown in Appendix A. 

### 4.3. PCR Analysis of VvmybA1 Alleles

Primers (listed in Appendix A) were purchased from Metabion (Planegg, Germany). PCR was performed in a final reaction volume of 25 µL containing 20–30 ng genomic DNA, 0.3 mM dNTPs, 1 × KAPA HiFi Buffer, 0.3 mM of each primer, and 0.5 U polymerase (KAPAHiFiTM Hot-Start PCR-Kit from Peqlab, Erlangen, Germany). A Mastercycler^®^ gradient (Eppendorf AG, Hamburg, Germany) was used for amplification with the following touchdown cycle program: initial activation at 95 °C for 5 min; followed by 10 cycles of 98 °C for 20 s, optimal annealing temperature −5 °C (specific for each PCR reaction; listed in Appendix A) for 15 s (+0.5 °C each cycle temperature increment), and 72 °C for 120 s; followed by 20 cycles of the same program with the optimal annealing temperature; then a final extension at 72 °C for 5 min. For visualization, DNA fragments were separated on a 1% agarose gel (Biozym Scientific GmbH, Hessisch Oldendorf, Germany) by electrophoresis.

### 4.4. Sequencing of BAC Clones of ‘Riesling Rot’

In order to display the complete mutated haplophase of ‘Riesling Rot’ at the grapevine berry color locus on chromosome 2, the respective genomic region was sequenced using BACs (bacterial artificial chromosomes).

To create a BAC library, potted plants were produced from the commercially used ‘Riesling Rot’ clone Gm4 (biological material ID: DEU098_VIVC10076_DEU098-1980-082) from woody cuttings in 2014 and cultivated in the greenhouse. About 70 g of young leaf mass (kept in the dark 24 h before harvest) were collected and shock-frozen in liquid nitrogen. The collected leaf samples were sent on dry ice to an external service provider (Genomics Institute at Clemson University, Clemson, SC, USA) to create the BAC library. After extraction of high-molecular DNA and fragmentation by a restriction enzyme, the approximately 90–140 kbp fragments were ligated into the linearized and dephosphorylated BAC vector pCUGIBAC1 [50]. This was followed by transformation into the *Escherichia coli* strain K12 DH10B (*F- endA1 recA1 galE15 galK16 nupG rpsL ΔlacX74 Φ80lacZΔM15 araD139 Δ(ara, leu)7697 mcrA Δ(mrr-hsdRMS-mcrBC) λ-*). Twenty-two pools, each with approximately 2000 CFU (Colony Forming Units; approximately 44,000 BAC clones), were created with a theoretical six-fold coverage of the genome. The BAC library screening scheme performed at Clemson University’s Genomics Institute is shown in Appendix A. Primer combination for Probe 1 served as detection for the berry color locus and the primer combination for Probe 2 specifically bound to the white haplophase via *Gret1*. The red haplophase could therefore be selected from a positive signal for Probe 1 and a negative signal for Probe 2. For this purpose, positive pools for both combinations were first selected by means of PCR, then plated out on a selective medium and each transferred to 384-well microtiter plates for DNA hybridization (single clone selection). Probes were prepared using the primer sequences corresponding to Probe 1 and Probe 2 based on the gDNA of ‘Riesling Rot’. In each case, several pools could be selected that clearly carried a fragment of the white (nine individual clones tested positive) and the red (six individual clones tested positive) haplophase. After completion of the screening, the BAC library (distributed on 48 384-well microtiter plates) and the identified positive individual clones were sent back to the Institute for Grapevine Breeding Geilweilerhof.

Since the screening with Probes 1 and 2 only served to detect the *VvmybA1* genomic region, a final check of the approximate size and position of the fragments in relation to the complete locus was carried out with SSR markers VVNTM1, GF02-61, GF02-62, GF02-68, VVNTM3, VVNTM5, VVNTM6 and GF02-58 (physical positions listed in Appendix A). Additionally, the BAC clones were tested for the presence of *VvmybA* genes and two clones with the largest inserts for both haplophases were selected. Sequencing was conducted at an external service provider (Seq-It GmbH and Co. KG, Kaiserslautern, Germany). A Shotgun library (Nextera XT) was created for each clone and sequenced using Illumina MiSeq (2 × 250 bp paired-end reads, approximately 100-fold coverage). Raw data were processed using the CLC Genomics Workbench (QIAGEN Bioinformatics, Aarhus, Denmark) at the Institute for Grapevine Breeding Geilweilerhof. For this purpose, the data were first trimmed (quality score: 0.05; ambiguous nucleotides: 2) and then a de novo assembly of all reads was carried out to get larger contigs. After completion of the assembly, residues of the BAC vector (pCUGIBAC1) were removed. Final sequences of both haplophases were deposited at GenBank database (http://www.ncbi.nlm.nih.gov/genbank accessed on: 28 February 2022) with the accession numbers OM885363 and OM885364.

## Figures and Tables

**Figure 1 ijms-23-03708-f001:**
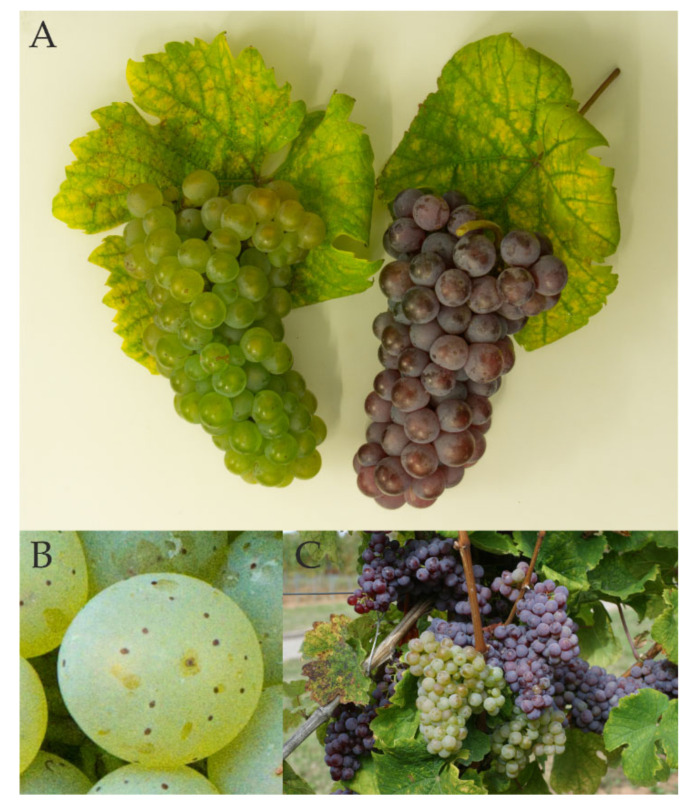
(**A**) Ripe grape bunches of ‘Riesling Weiss’ and ‘Riesling Rot’. (**B**) Ripe ‘Riesling Weiss’ berry with clearly visible lenticels. (**C**) ‘Riesling Rot’ with the typical red bunches and two mutated white bunches on a single fruit cane.

**Figure 2 ijms-23-03708-f002:**
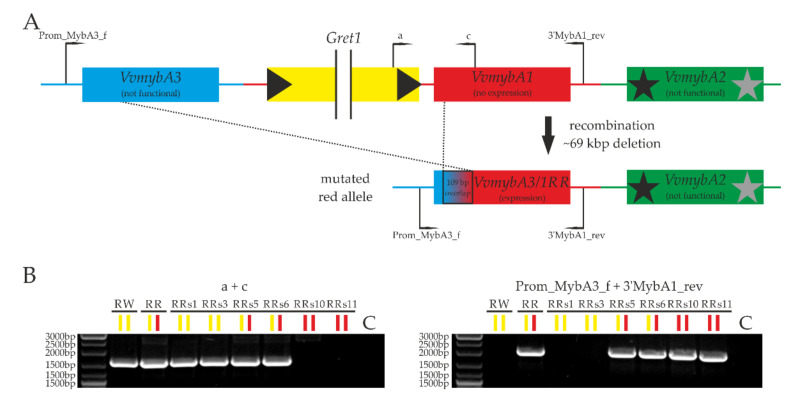
The unique mutation of ‘Riesling Rot’ at the grapevine berry color locus on chromosome 2. (**A**) Schematic overview of the two ‘Riesling Rot’ haplophases at the BCL based on BAC clone sequencing. Primer binding sites used in (**B**) are indicated. Dotted lines represent the putative recombination site. (**B**) PCR results for ‘Riesling Weiss’, ‘Riesling Rot’, and the ‘Riesling Rot’ selfing line RRs. Colored bars represent the haplotype constitution at the berry color locus (yellow = white allele, red = red allele). RW = ‘Riesling Weiss’; RR = ‘Riesling Rot’; C = no template control.

**Figure 3 ijms-23-03708-f003:**
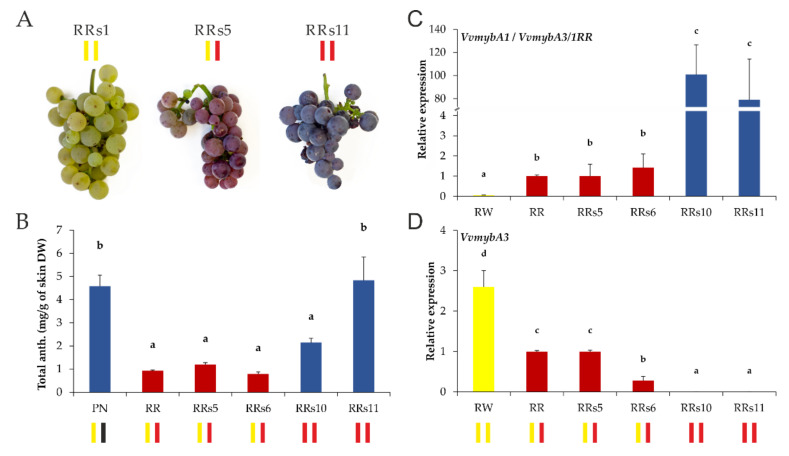
Total anthocyanin content and relative gene expression of *VvmybA1*/*VvmybA3/1RR* and *VvmybA3* of ‘Riesling Weiss’, ‘Riesling Rot’, and the self-cross line RRs in ripe berry skins. (**A**) Ripe bunches of the genotypes RRs1, RRs5, and RRs11 with differing haplotype constitutions at the berry color locus. (**B**) Total anthocyanin content in ripe berries. The classical red wine cultivar ‘Pinot Noir’ with dark blue berries is used as a reference. Data represent the mean values of three independent replicates; error bars represent standard deviation. Relative gene expression of *VvmybA1*/*VvmybA3/1RR* (**C**) and *VvmybA3* (**D**). Data represent the mean values of three independent replicates with normalization to the expression of RR (RW and RR) and RRs5 (RRs line); error bars represent standard errors. Colored bars represent the haplotype constitution at the berry color locus (yellow = white allele, red = red allele, black = wild type blue/black allele of typical red wine cultivars with functional *VvmybA1* and *VvmybA2*). Different letters indicate statistically significant differences (*p*-value < 0.05). PN = ‘Pinot Noir’; RR = ‘Riesling Rot’; DW = dry weight.

**Figure 4 ijms-23-03708-f004:**
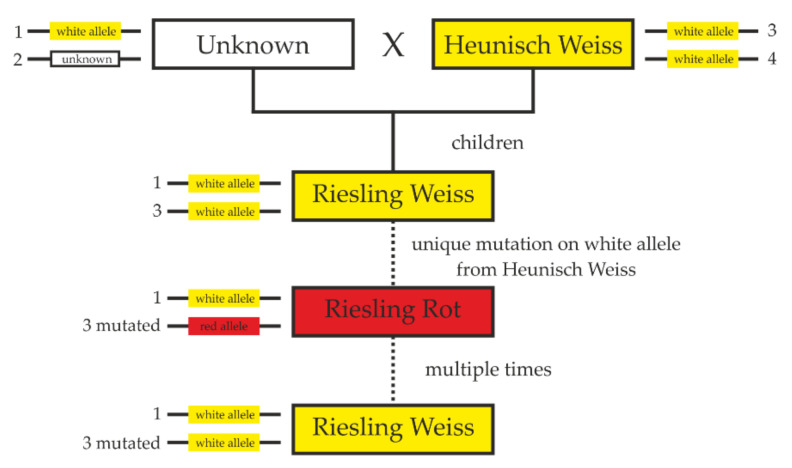
Pedigree tree of ‘Riesling’ based on parental haplophase assignment using SSR marker data along chromosome 2.

## Data Availability

The BAC sequence data have been deposited in the GenBank database (http://www.ncbi.nlm.nih.gov/genbank accessed on: 28 February 2022) with the accession numbers OM885364 (BAC07A05) and OM885363 (BAC05I18).

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
