# Peer review of "A 69 kbp Deletion at the Berry Color Locus Is Responsible for Berry Color Recovery in Vitis vinifera L. Cultivar ‘Riesling Rot’"

_ijms, 2022, doi:10.3390/ijms23073708_

Round 1
Reviewer 1 Report
Röckel and colleagues present an interesting study on discover the molecular reason of color recovery of Riesling Rot from red to white. they first create self lines (homozygous) to make it easy to detected by SSR markers. Overall, this manuscript is well designed and presented with clear and interesting findings. I recommend to publish after authors address the following minor comments:
- Put Tables in order from Table S1 to Table S8.
Introduction
- In the introduction part, authors gave a detailed description of the background, somehow, which is not concise enough for the article, but more suitable for a book.
- And some large paragraphs are missing references, such as Line 66-70.
- Some of these descriptions are not based on the results of scientific research, but elsewhere, such as Line 49-50.
Results
- 6, 12 and 8 were selected according to what criteria, I can't tell from Table S2. are they all self line? or homozygous? seems not. and how many seedling did you used totally? Actually, I don't know what you were trying to say in the paragraph 2.1.
6. Please explain how the conclusions in Figure 4 were drawn from Table S7.

Author Response
Thank you for your comprehensive review with many insightful comments to improve our work. Our Responses to your remarks are marked in bold.
Röckel and colleagues present an interesting study on discover the molecular reason of color recovery of Riesling Rot from red to white. they first create self lines (homozygous) to make it easy to detected by SSR markers. Overall, this manuscript is well designed and presented with clear and interesting findings. I recommend to publish after authors address the following minor comments:
- Put Tables in order from Table S1 to Table S8.
Thank you for the finding. We carefully checked the manuscript and changed the ordering of the Tables S1 to S8. Table S3 is now Table S8 and Table S4-8 decreased by one, respectively.
Introduction
- In the introduction part, authors gave a detailed description of the background, somehow, which is not concise enough for the article, but more suitable for a book.
We carefully read the introduction again and concluded that this comment refers to the section about the historical background of ‘Riesling’. Therefore, we shortened the paragraph (originally lines 81-110) about 2 long sentences to be more concise for a research article. It should fit better now.
- And some large paragraphs are missing references, such as Line 66-70.
Reference added.
- Some of these descriptions are not based on the results of scientific research, but elsewhere, such as Line 49-50.
Reference added.
Results
- 6, 12 and 8 were selected according to what criteria, I can't tell from Table S2. are they all self line? or homozygous? seems not. and how many seedling did you used totally? Actually, I don't know what you were trying to say in the paragraph 2.1.
The main purpose of paragraph 2.1 is to explain why and how we created the self-crossed lines. We extensively reworked paragraph 2.1 and addressed all questions/remarks to facilitate the understanding. Additionally, we added a column in Table S2 with the haplophase status to further increase understandability.
- Please explain how the conclusions in Figure 4 were drawn from Table S7.
We reworked paragraph 2.4 to clarify the conclusions: “Based on the BCL-specific genetic profile of ‘Heunisch Weiss’, the inherited ‘Riesling Rot’ haplophases could be assigned to the respective parents (Table S7). It became evident that the mutation-bearing haplophase (possesses the VvmybA3/1RR allele) originates from the haplophase originally passed on by ‘Heunisch Weiss’. Consequently, the white variant ‘Riesling Weiss’ must have been first and mutated to red, since no color mutant is described for the parent variety ‘Heunisch Weiss’ that could have passed the mutation.”. Additionally, we reworked Table S6 to further increase understandability.
Best regards
Reviewer 2 Report
The manuscript is very well drafted and the results are presented clearly. There is one minor concern: Please increase the labels of the axes in figure 3 as they are currently hard to read.
Author Response
Thank you for your Review. We increased the label sizes of the axes in Figure 3.